# Carvacrol-Induced Vacuole Dysfunction and Morphological Consequences in *Nakaseomyces glabratus* and *Candida albicans*

**DOI:** 10.3390/microorganisms11122915

**Published:** 2023-12-04

**Authors:** Eliz Acuna, Easter Ndlovu, Ali Molaeitabari, Zinnat Shahina, Tanya Elizabeth Susan Dahms

**Affiliations:** Department of Chemistry and Biochemistry, University of Regina, Regina, SK S4S 1P4, Canada; elizzyraa@gmail.com (E.A.);

**Keywords:** carvacrol, hyphae, pH, membrane integrity, morphogenesis, vacuole

## Abstract

With the prevalence of systemic fungal infections caused by *Candida albicans* and non-*albicans* species and their resistance to classical antifungals, there is an urgent need to explore alternatives. Herein, we evaluate the impact of the monoterpene carvacrol, a major component of oregano and thyme oils, on clinical and laboratory strains of *C. albicans* and *Nakaseomyces glabratus*. Carvacrol induces a wide range of antifungal effects, including the inhibition of growth and hyphal and biofilm formation. Using biochemical and microscopic approaches, we elucidate carvacrol-induced hyphal inhibition. The significantly reduced survival rates following exposure to carvacrol were accompanied by dose-dependent vacuolar acidification, disrupted membrane integrity, and aberrant morphology. Germ tube assays, used to elucidate the relationship between vacuolar dysfunction and hyphal inhibition, showed that carvacrol significantly reduced hyphal formation, which was accompanied by a defective *C. albicans* morphology. Thus, we show a link between vacuolar acidification/disrupted vacuole membrane integrity and compromised candidal morphology/morphogenesis, demonstrating that carvacrol exerts its anti-hyphal activity by altering vacuole integrity.

## 1. Introduction

*Candida* are opportunistic pathogenic fungi that reside as commensal organisms in the human oral mucosa, skin, and gastrointestinal tracts of healthy individuals [1,2]. Damage to epithelial barriers, dysbiosis of resident microbiota, and the dysfunction of the host’s immune system favor conversion from the commensal yeast into a pathogen that is capable of causing infection [3,4]. *Candida* causes superficial cutaneous and mucosal infections and systemic infections termed candidiasis, with risk factors including immune suppression and immunodeficiency resulting from chemotherapy or radiotherapy for cancer treatment, human immunodeficiency virus, surgical procedures, long-term hospitalization, and medical devices such as catheters, to name a few [3,4,5,6,7,8]. The spread of infection through the bloodstream to other organs such as the brain, kidneys, lungs, and liver, namely invasive or systemic candidiasis, is associated with a yearly mortality rate in hospitals as high as 40–60% [4,6,9].

*C. albicans* and *Nakaseomyces glabratus* (previously known as *C. glabrata*) are the most common pathogenic yeasts, with the latter being the most clinically isolated species and prevalent cause of candidiasis [10,11]. *C. albicans* and *N. glabratus* have a number of key differences. For example, *C. albicans* is diploid and can morphologically switch from yeast to hyphae (dimorphic), whereas *N. glabratus* is haploid and not dimorphic [8,12]. In terms of epidemiology, *N. glabratus* is the second cause of *Candida* infections after *C. albicans*, the latter having a greater number of virulence factors [12]. One of the key features allowing *C. albicans* to thrive as a successful pathogen is its ability to survive under different host nutrient availability, CO_2_, O_2_, and pH levels [3,6]. For example, the host tissue environment, including the temperature (37 °C), serum, neutral pH, and hormones, can induce *C. albicans*’ yeast–hyphal morphological switching [2,3], considered a virulence trait along with biofilm formation, the evasion of the host’s immune system, adhesion to host surfaces, and the secretion of toxins (e.g., candidalysin) and hydrolytic enzymes (e.g., aspartyl proteases) [2,3]. *C. albicans*’ virulence traits play crucial roles in its pathogenesis and are therefore promising targets for vaccine and antifungal drug development [4].

At present, the treatment of candidiasis mainly involves the use of four antifungal therapeutics, namely, azoles, polyenes, echinocandins, and flucytosine [6,13,14]. Antifungal resistance is a challenge in treating invasive candidiasis [1,4] necessitating the search for other compounds with different mechanisms that represent novel antifungal strategies. Essential oils extracted from plants have emerged as antifungal alternatives [15] with their low cost, broad spectrum of activities, fewer side effects, and lower toxicity [16]. Numerous studies provide support for the therapeutic efficacy of essential oils and their components against a wide range of fungal infections, comparable to classic fungicides [16,17,18,19,20]. These compounds and mixtures mainly exert their antifungal effect by altering cell membrane fluidity, thereby altering the permeability of the mitochondrial membrane [20]. Thyme, one such essential oil studied for its antifungal activities, has the major components thymol and carvacrol [15]. Carvacrol is a monoterpene phenol and a potent antifungal against multiple clinical strains of *C. albicans* [19,21,22]. Although the definitive antifungal mechanism of carvacrol remains unclear, its toxicity has been linked to the disruption of the cell membrane, ergosterol production, endoplasmic reticulum stress, and H^+^ and Ca^2+^ dyshomeostasis [19,21,22,23,24].

Vacuoles, the largest fungal organelles, play a role in many crucial cellular processes that maintain homeostasis, including responses to environmental stresses and morphological switching (the yeast–hyphal transition) [5,25]. Further, vacuolar ATPases (V-ATPases) regulate ion homeostasis by driving the translocation of protons across the vacuolar membrane into the lumen [14] to maintain an acidic pH [25,26]. Azole treatment can perturb vacuole morphology [14] and decrease vacuolar acidification in *N. glabratus* [25], and V-ATPase inhibition interferes with *C. albicans*’ morphological switching by reducing nutrient acquisition [27]. V-ATPase dysfunction may interfere with cellular processes that trigger fungal infections, such as filamentation and drug tolerance, making it an attractive target for drug discovery.

This study examines carvacrol as an antifungal against *C. albicans* and *N. glabratus* and its impact on vacuoles. Our findings validate the importance of essential oils as alternative antifungal agents and V-ATPase as a drug target.

## 2. Materials and Methods

### 2.1. Strains and Growth Conditions

The *C. albicans* lab strain was procured from the ABC Platform Bugs Bank, Nancy, France [28], *N. glabratus* KUE100 was kindly provided by Prof. Miguel Teixeira (Universidade de Lisboa, Lisbon, Portugal), and *C. albicans* and *N. glabratus* clinical isolates were obtained from blood cultures through the Regina Qu’Appelle Health Region (Department of Microbiology, Regina General Hospital, Regina, SK, Canada). These strains were stored at −80 °C in 50% glycerol before being revived on 1% yeast extract–2% peptone–2% dextrose (YPD) agar plates for 24 h at 30 °C. A single colony was then selected from the agar plate, inoculated into YPD broth, and grown overnight for 18 h at 30 °C in a shaking incubator operating at 200 rpm.

### 2.2. Minimum Inhibitory Concentration (MIC)

The MIC values for carvacrol were determined using the broth microdilution method that was adapted from the M27-A3 method, as outlined by the Clinical and Laboratory Standards Institute (CLSI) [29]. Briefly, working solutions of carvacrol (Sigma; W224502), 800 μg/mL (lab strains) and 1000 μg/mL (clinical strains), were prepared by adding 0.82 μL or 1.02 μL of 98% carvacrol to YPD containing 1% dimethylsulfoxide (DMSO) to a final volume of 1 mL. The working solution (100 μL) was then added to a 96-well microtiter plate (Sardstedt, Nümbrecht, Germany), after which it was serially diluted with YPD containing 1% DMSO. Cultures, diluted to an optical density at 600 nm (OD_600_) of 0.001, corresponding to approximately 1 × 10^5^ cells/mL, were added to each well (100 µL). Unless otherwise stated, assay blanks consisted of carvacrol only in YPD, and growth controls were yeast cultures in the absence of carvacrol. The microtiter plate was sealed with parafilm to prevent the evaporation or diffusion of carvacrol into other wells, incubated for 24 h at 30 °C with shaking (200 rpm), and the OD_600_ measured using a plate reader (BioTek; Winooski, VT, USA). The MIC was determined as the lowest concentration of carvacrol to eliminate growth.

### 2.3. Vacuolar Acidification

Since vacuoles play a vital role in the yeast–hyphal transition, antifungal resistance, and virulence [25], their response to carvacrol was assessed. The use of a neutral red stain that preferentially accumulates within the low-pH environment of the vacuolar lumen followed a previous study [30] with slight modifications. Briefly, a carvacrol (500 μL) working solution (2 × MIC) was serially diluted in YPD containing 1% DMSO in a 24-well microtiter plate (Sardstedt, Nümbrecht, Germany). Strains grown to the mid-log phase were diluted to approximately 1 × 10^7^ CFU/mL, 500 μL was added to each well, the plate was sealed and incubated for 4 h at 30 °C with shaking (200 rpm). Cells were then harvested via centrifugation at 5000× *g* for 5 min, washed thrice in 0.001 M phosphate-buffered saline (PBS, 0.138 M NaCl, 0.0027 M KCl, pH 7.4), and resuspended in 100 μL of 0.001 M PBS, and the OD_600_ was measured. To normalize the cell number, the growth control and cells exposed to carvacrol at 1/4 and 1/2 MIC were diluted to the same OD_600_ as those exposed to MIC. A 25 μL aliquot of either a carvacrol-treated or untreated cell suspension was mixed with 5 μL of a neutral red stain (2 mg/mL (*w*/*v*); Sigma), incubated for 5 min at room temperature (RT), and visualized using bright-field microscopy (Olympus BX51). Cells were quantified as the percentage of intact, partially perturbed, and perturbed vacuoles from a total of 100 individual cells captured in ten microscopic fields of view (FOVs) for each of three biological replicates.

### 2.4. Membrane Integrity Assay

To assess the impact of carvacrol on *C. albicans*’ and *N. glabratus*’ vacuole membrane integrity, the lipophilic yeast vacuole membrane-specific dye MDY-64 (Invitrogen, ThermoFisher Scientific; Saskatoon, SK, Canada; 10799033) was monitored (λ_ex_ = 451 nm; λ_em_ = 497 nm). Cell suspensions were prepared as above, and the cells were stained according to the manufacturer’s recommendations with a slight modification. Briefly, cells resuspended in 100 μL of PBS were mixed with a 100 μM MDY-64 stock solution made in 1% DMSO to a final concentration of 10 μM. The samples were then incubated (RT, 3 min) in the dark, and microscopic slides were prepared and visualized using epifluorescence microscopy (Carl Zeiss Axio Observer Z1 inverted microscope, Oberkochen, Germany) at 63× under oil immersion. The results were expressed as the percentage of cells having intact, partially ruptured, and ruptured vacuole membranes, calculated from a total of 100 individual cells captured in 10 FOVs for each of three biological replicates.

### 2.5. Germ Tube Formation Assay

The effect of carvacrol on *C. albicans*’ germ tube formation and on *N. glabratus* was assessed according to established protocols [31] with modifications. Briefly, overnight *Candida* cultures in the mid-logarithmic phase were diluted to approximately 1 × 10^7^ CFU/mL in prewarmed YPD with 1% DMSO and 10% fetal bovine serum (FBS) and added to 24-well microtiter plates (Sardstedt, Nümbrecht, Germany) containing 500 μL of carvacrol serially diluted to 1/2 of the MIC and 1/4 of the MIC. *Candida* strains incubated in only YPD with 1% DMSO and 10% FBS were used as growth controls. After 4 h of incubation at 37 °C with shaking at 200 rpm, the cells were centrifuged at 5000× *g* for 5 min, followed by three washes with PBS. Microscopy slides were then prepared using 5 μL of the cell suspension and examined at 50× using the bright field configuration of a fluorescence microscope (Olympus BX51 fluorescence microscope, Olympus Co., Ltd., Richmond Hill, ON, Canada). In addition to observing the vacuole conditions during induced *C. albicans* hyphal growth, controls were stained with neutral red and MDY-64 as described above. The percentages of fission yeast, chain-forming cells, germ tubes, pseudohyphae, and hyphae were determined from 100 cells of 10 microscopic FOVs for each of three biological replicates.

### 2.6. Statistical Analysis

GraphPad Prism^®^ Software (version 9, GraphPad Software, Inc., La Jolla, CA, USA) generated graphical representations and statistical analyses. A one-way analysis of variance (ANOVA) was used with Dunnett’s multiple comparison post-test to compare all test data to controls. Statistically significant differences of *p* < 0.05 (*), *p* < 0.01 (**), *p* < 0.001 (***), or *p* < 0.0001 (****) are indicated in figures, and the absence of an asterisk indicates no statistical difference (*p* > 0.05, ns). All assays were conducted in triplicate for each of three biological replicates, and error bars represent the standard error of the mean (±SEM).

## 3. Results

### 3.1. C. albicans and N. glabratus Growth Was Similarly Inhibited by Carvacrol

The growth of *C. albicans* and *N. glabratus* lab strains (Figure 1a) was completely inhibited (MIC) by 200 μg/mL carvacrol, and the clinical strains (Figure 1b) by 250 μg/mL carvacrol. Interestingly, the growth of the lab strains at 1/2 of the MIC (100 μg/mL) was similar to that observed at the MIC, with approximately 5% growth compared to the control (Figure 1a). On the other hand, the clinical isolates at half of the MIC (125 μg/mL) had significantly higher growth (Figure 1b). Carvacrol also altered the overall morphology of all strains (Appendix A).

### 3.2. Carvacrol Disrupts Vacuole Acidification in C. albicans and N. glabratus

A vacuole acidification assay was used to investigate carvacrol-induced damage to the vacuole, specifically its acidification, which is essential for intracellular ion homeostasis in *C. albicans* and *N. glabratus* [25,32]. Figure 2 shows a dose-dependent effect of carvacrol on vacuole morphology. Cells were characterized (Figure 2b,c) as either intact (the retention of the neutral red stain), partially perturbed (visibly reduced vacuolar staining), or perturbed (completely unstained or leaky vacuoles). For all strains, 80–90% of the control cells had intact vacuoles, with approximately 15% having either partially perturbed or perturbed vacuoles (Figure 2a,c). Carvacrol treatment resulted in a higher number of vacuoles with a perturbed pH directly proportional to the oil concentration (from 1/4 to full MIC), but with some variability in the degree of perturbation (Figure 2a,c). Compared to the controls, the vacuoles of carvacrol-treated *C. albicans* lab and clinical strains were significantly damaged at 1/4 of the MIC, 1/2 of the MIC and the MIC based on a significant decrease in cells with intact vacuoles (Figure 2a,c). There were some differences in vacuole and cell morphology in response to carvacrol between the clinical and lab strains. It is important to note that at 1/4 of the MIC (Figure 2a), while the vacuoles of the lab strain remained unstained and intact, their clinical counterparts had a stained cytoplasm associated with vacuolar leakage.

The *N. glabratus* lab strain was significantly impacted by exposure to lethal concentrations of carvacrol, as indicated by the number of perturbed (50%) and partially perturbed (15%) vacuoles (Figure 2c). Interestingly, strains exposed to carvacrol at 1/4 and 1/2 of the MIC had no significant increase in the number of partially perturbed vacuoles compared to the control (Figure 2c); however, there was a significant increase in the number of perturbed vacuoles (70%). The clinical strains showed a significant increase in the number of perturbed and partially perturbed vacuoles, both at lethal and sub-lethal carvacrol concentrations (Figure 2c), and this was accompanied by vacuolar leakage (cytoplasmic staining; Figure 2a).

### 3.3. Carvacrol Disrupts C. albicans and N. glabratus Vacuolar Membrane Integrity

Based on the impact of carvacrol on *Candida* vacuoles, vacuole membrane integrity was assessed using the dye MDY-64. Cells were characterized (Figure 3b) as either intact (fully stained vacuolar membrane on rounded vacuoles), partially ruptured (visibly reduced vacuole membrane staining), or ruptured (stain distributed throughout the cell). Vacuole morphology, assessed via fluorescence microscopy, revealed that cells with ruptured membranes also had fragmented vacuoles and that the number proportionally increased as a function of carvacrol concentration for all *Candida* strains (Figure 3a,c). At the MIC, almost 80% of the vacuolar membranes were ruptured, with very few cells having intact vacuolar membranes (Figure 3c).

These results are similar to those of the vacuole acidification assay (Figure 2), suggesting that there is indeed a dose-dependent effect on vacuole membrane morphology (Figure 3c). Furthermore, we noted that there was a decrease in the overall cell size for carvacrol-treated cells compared to untreated controls (Figure 3a). Both *C. albicans* and *N. glabratus* revealed an abundance of chain-forming cells when exposed to carvacrol at 1/4 of the MIC as compared to the control, 1/2 of the MIC, and the MIC (Figure 3a).

### 3.4. Carvacrol Disrupts Hyphal Formation in C. albicans

Given the vacuolar disruption by carvacrol for *C. albicans* and *N. glabratus*, and the role of vacuole-regulated cell expansion and ion homeostasis in virulence, the impact of carvacrol on *C. albicans*’ hyphal formation was investigated [33]. Using a serum-induced assay and microscopic imaging, we quantified the ratio of yeast to hyphal morphology and the number of germ tubes and pseudohyphae (precursors to hyphae). *C. albicans* treated with lethal and sublethal concentrations of carvacrol had reduced germ tubes, pseudohyphae, and hyphae as compared to controls (Figure 4a—left panel). At 1/4 of the MIC, chain-forming cells were most abundant, contrary to 1/2 of the MIC and the MIC, for which most of the cells appear as budding yeast (80%) with only 20 and 10% chain-forming cells, respectively (Figure 4b—left panel). At 1/4 of the MIC there was a statistically significant increase in the number of pseudohyphae compared to the control; however, pseudohyphal formation was inhibited by carvacrol at 1/2 of the MIC and at the MIC. Overall, we observe a dose-dependent inhibition of *C. albicans* lab strain hyphal formation by carvacrol (Figure 4—left panel).

On the other hand, carvacrol did not completely inhibit germ tube and hyphal formation for the *C. albicans* clinical isolate, even at the MIC and despite a significant reduction (<5%) at 1/2 of the MIC and the MIC (Figure 4b—right panel), with increased pseudohyphae only at 1/4 of the MIC and 1/2 of the MIC compared to the control (Figure 4b—right panel). Interestingly, at 1/4 of the MIC, most of the cells appeared as germ tubes, at 1/2 of the MIC, most were budding yeast, and at the MIC, most were chain-forming yeast (Figure 4b—right panel). These results are in contrast with the lab isolate (Figure 4b—left panel), for which approximately 100% of the cells were budding yeast with carvacrol treatment at the MIC.

To account for potential changes in the response to vacuole expansion during hyphal formation and for possible temperature-induced changes, vacuole acidification and membrane integrity were assessed simultaneously. Untreated cells had intact and, for the most part, acidified vacuoles, whereas the carvacrol-treated cells showed dose-dependent vacuole membrane rupture, with condensed vacuoles consistently observed at the MIC (Figure 4a), consistent with the vacuole assay (Figure 3).

### 3.5. Carvacrol Induces Chain-Forming Cells in N. glabratus

Unlike *C. albicans*, *N. glabratus* lacks the ability to form true hyphae yet can cause lethal infections. Untreated *N. glabratus* (Figure 5a) had no germ tube, pseudohyphae, or hyphae, as expected, but when treated with carvacrol (Figure 5b), both *N. glabratus* strains had chain-forming cells. Changes in vacuole morphology were also consistent with the vacuole assays (Figure 3) in which cells exposed to carvacrol at 1/2 MIC and MIC had visibly reduced cell and vacuolar sizes which are potentially attributed to a condensed vacuolar morphology (Figure 5a). Indeed, there was a dose-dependent decrease in the number of cells with intact vacuolar membranes and vacuolar acidification (Figure 5b).

## 4. Discussion

The fungal vacuole is a highly complex organelle involved in functions such as the maintenance of cellular homeostasis and the regulation of ions, basic amino acid concentrations, and the intracellular pH [34,35]. Therefore, changes in ion homeostasis result in impaired growth and subsequent cell death [36]. Vacuolar size thresholds are required for cell cycle progression during hyphal growth [37]; thus, yeast vacuoles have been the focus of searches for plausible cellular mechanisms of antifungal toxicity in *C. albicans* [38,39,40] but not in the context of essential oil components. In this study, we characterized carvacrol-mediated vacuolar defects in *C. albicans* and *N. glabratus* and for *C. albicans* in relation to their growth and hyphal inhibition, which can also be linked to the regulation of branching frequency. Carvacrol was lethal for both *C. albicans* and *N. glabratus* at concentrations (Figure 1) consistent with previously published data [21,41,42,43]. Clinical isolates are typically more pathogenic based on the production of proteases, hemolysins, and esterases [44], consistent with the higher MIC required to inhibit their growth in comparison to nonclinical strains (Figure 1).

The maintenance of an acidic vacuolar environment is an integral factor in their versatile role supporting yeast proliferation during fungal infection [34]. Our data show a dose-dependent alkalinization of the vacuole with carvacrol treatment, implying the leakage of protons from the vacuole to the cytosol and its acidification [36], confirmed via neutral red staining (Figure 2). Vacuolar proton leakage may result from the disruption of V-ATPase, the proton antiporter responsible for intracellular ion homeostasis. Indeed, carvacrol disrupts V-ATPase in *S. cerevisiae* [24,36], consistent with vacuolar membrane disruption (Figure 3) and likely attributed to carvacrol lipophilicity since V-ATPase is a vacuolar-membrane-bound transporter. The lab strains exposed to carvacrol at 1/4 of the MIC remained intact (Figure 2a) while their clinical counterparts showed vacuolar leakage, suggesting that other factors in addition to perturbed vacuolar membranes contribute to vacuolar alkalinization. With exposure to carvacrol at a sublethal MIC, the lab strains evaded toxicity and cell death by mitigating ion leakage from the vacuole into the cytoplasm, consistent with previous reports for carvacrol, clove, and thyme [45]. Likewise, vacuolar membrane integrity (Figure 3) was perturbed in a dose-dependent manner as in prior studies on carvacrol [24,36], other essential oils (honokiol), and conventional antifungal drugs such as amphotericin B [5,46]. The combined disruption of vacuolar pH and membrane integrity, integral for the vacuole-mediated regulation of osmotic pressure and yeast cell volume, largely through ion and membrane trafficking, is consistent with the observed phenotypes and increased sensitivity of these strains to osmotic imbalances upon treatment with carvacrol [24,36]. Thus, ion dyshomeostasis combined with increased membrane permeability results in these phenotypes. An abundance of chain-forming cells at sublethal levels of carvacrol (1/4 of the MIC; Figure 3a) is novel, and it is interesting to speculate an adaptation of morphology during antifungal resistance, similar to the mechanism observed in *C. auris* [47]. Interestingly, reductions in cell-separation enzymes, increased cell wall chitin, and stress-induced inhibition in cell separation are linked to membrane and microtubule disruption, leading to pseudohyphal/chain formation, which prevents cell separation [48]. Carvacrol interferes with ergosterol biosynthesis, ultimately affecting cell membrane integrity [23], a potential cause of increased chain formation meriting further investigation.

Hyphal formation, which contributes to adhesion with and the invasion of host cells, is a key virulence factor for *C. albicans*, enabling it to invade and infect host cells. On the other hand, *N. glabratus* is virulent yet unable form hyphae, which is perhaps why *C. albicans* is more virulent. Several recent studies of *C. albicans* established the vital role played by the fungal vacuole in supporting hyphal formation and the establishment of infection [49,50,51], with vacuolar ion homeostasis implicated not only in proliferation but also virulence and dimorphic switching [40,49]. Only a portion of the tubular vacuoles were acidified (Figure 4a), which may relate to the ion trafficking required for vacuolar expansion during hyphal elongation [51,52]. The inhibition of morphological switching by carvacrol is novel but consistent with the impact of oregano oil (high in carvacrol) on *C. albicans* biofilm formation, for which hyphal formation plays a major role [53]. The dose-dependent inhibition of *C. albicans* and *N. glabratus* by carvacrol (Figure 4b and Figure 5) is consistent with the impact of other essential oils, their components, and conventional antifungal drugs [45,47,54].

We recently reported that rosemary oil, eugenol, and citral [18,48] disrupt the vacuolar membrane, causing defects in vacuolar function and cell cycle inhibition, ultimately hindering hyphal progression. The concurrent carvacrol-induced disruption of vacuolar membrane integrity suggests a role for vacuole function in the carvacrol-induced inhibition of hyphal growth, ultimately reducing *C. albicans* virulence. Consistent with this idea, *C. albicans* mutants with deficient vacuoles have profound defects in invasive hyphal growth [34,46,55]. We propose that perturbations in vacuolar pH and membrane integrity disrupt vacuolar expansion and the delivery of essential metabolites required for hyphal growth, with partially inhibited cells having intact vacuolar membranes but an altered vacuolar pH (Figure 4a). Such an impact is similar to that for *Candida* spp. with mutated V-ATPase subunits known to have regulatory roles related to these vacuolar functions [25,37,56].

## 5. Conclusions

This study shows the impact of carvacrol on *C. albicans*’ and *N. glabratus*’ vacuolar morphology. Carvacrol exposure led to reduced vacuolar membrane integrity accompanied by significant vacuolar alkalinization, which compromised overall vacuolar morphology, particularly at lethal concentrations. Changes to vacuolar pH and morphology were dose-dependent, with the severity of the damage increasing as a function of concentration. The complete inhibition of *C. albicans* hyphal formation in the lab isolate and incomplete, yet significantly reduced, hyphal formation in the clinical strain highlight the impact of carvacrol on this virulence trait, along with the crucial hyphal precursors: germ tubes and pseudohyphae. *N. glabratus* exposed to carvacrol had a greatly reduced number of chain-forming cells, a feature potentially relevant to biofilm formation. We anticipate that compounds disrupting normal vacuolar function should compromise pathogenicity through sensitizing the fungal cells to physiological stress and inhibiting hyphal growth, a key determinant of *C. albicans* pathogenicity.

## Figures and Tables

**Figure 1 microorganisms-11-02915-f001:**
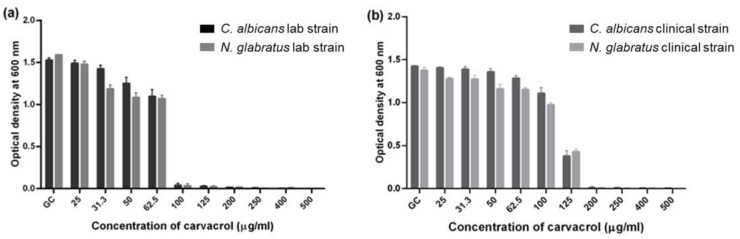
**Minimum Inhibitory Concentration (MIC) assay of carvacrol against *C. albicans* and *N. glabratus***. Overnight cultures of *Candida* (**a**) lab and (**b**) clinical strains exposed to carvacrol and incubated for 24 h at 30 °C. GC = growth control. Results are the means of 3 biological replicates.

**Figure 2 microorganisms-11-02915-f002:**
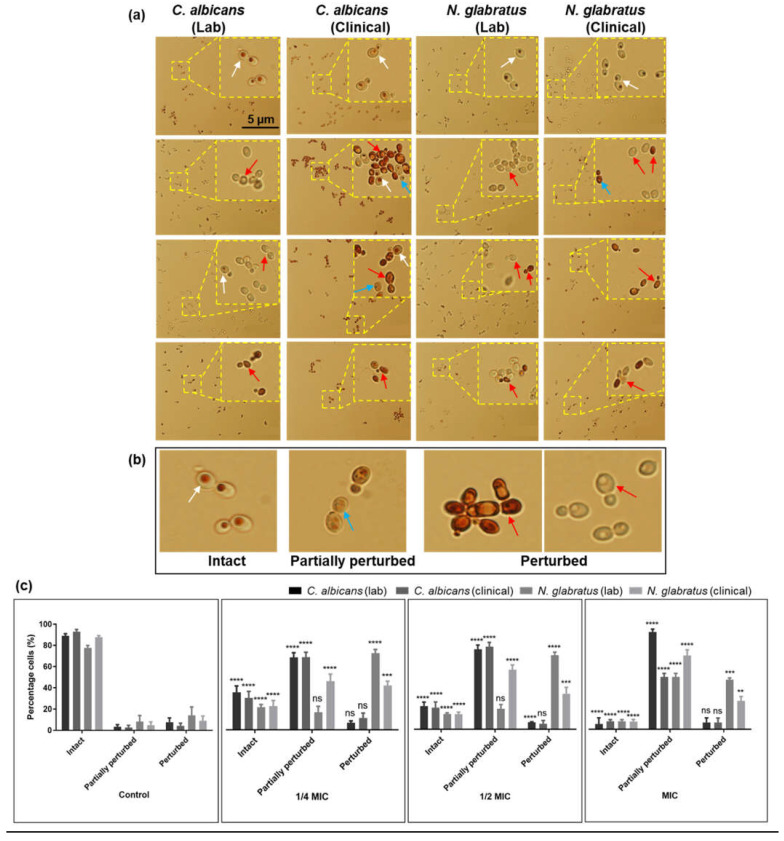
**Carvacrol induces vacuolar acidification in *C. albicans* and *N. glabratus*.** (**a**) Bright-field images of neutral-red-stained cells showed the impact of carvacrol on vacuoles. Scale bars are 5 μm and applicable to all images. (**b**) The white, blue, and red arrows indicate intact, partially perturbed, or fully perturbed vacuoles, respectively. (**c**) Bar graphs show the percentages (%) of cells with intact, partially perturbed, and perturbed vacuoles. Statistical significance (****, *p* < 0.0001; ***, *p* < 0.001; **, *p* < 0.01, ns (no significance), *p* > 0.05) was evaluated using a one-way ANOVA, followed by Dunnett’s multiple comparison test of each condition versus a control from three biological replicates captured from ten microscopic FOVs at 100×.

**Figure 3 microorganisms-11-02915-f003:**
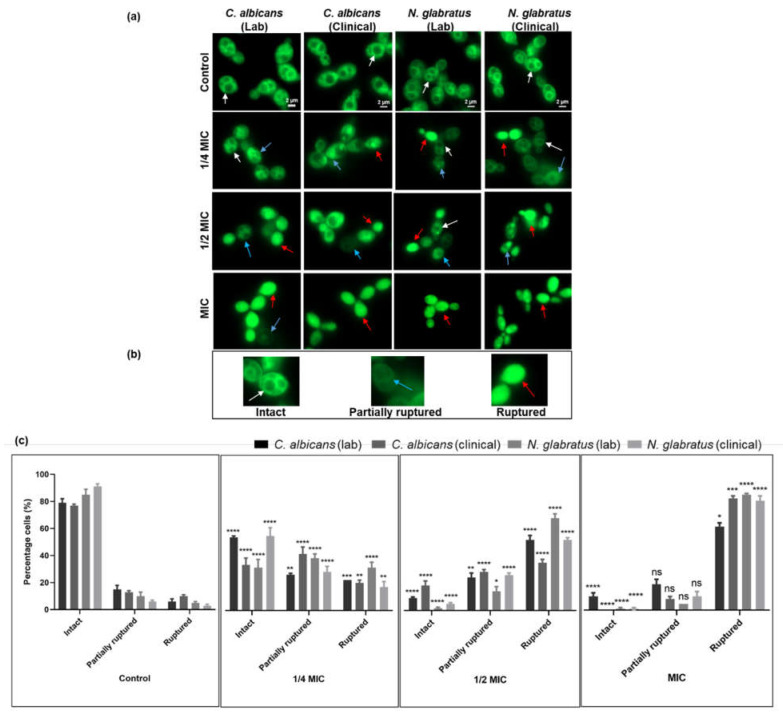
**Carvacrol disrupts vacuole membrane integrity in *C. albicans* and *N. glabratus*.** (**a**) Representative epifluorescence images of MDY-64 (λ_ex_ = 451 nm; λ_em_ = 497 nm) show the impact on vacuolar membranes after 4 h of carvacrol treatment. The scale bar for the control is 2 μm and represents the scale for all images. (**b**) The white, blue, and red arrows represent intact, partially ruptured, and ruptured vacuolar membranes, respectively. Lysed cells (red arrow) were observed mostly for strains treated with carvacrol at half of the MIC and the MIC. (**c**) Bar graphs show the percentage (%) of cells with intact, partially ruptured, and ruptured vacuole membranes. Data are reported as the means ± SEMs of three biological replicates, with 300 cells captured from ten microscopic FOVs (63×), for which statistical significance (****, *p* < 0.0001; ***, *p* < 0.001; **, *p* < 0.01, *, *p* < 0.05, ns (no significance), *p* > 0.05) was evaluated using a one-way ANOVA, followed by Dunnett’s multiple comparison test of each condition versus the control.

**Figure 4 microorganisms-11-02915-f004:**
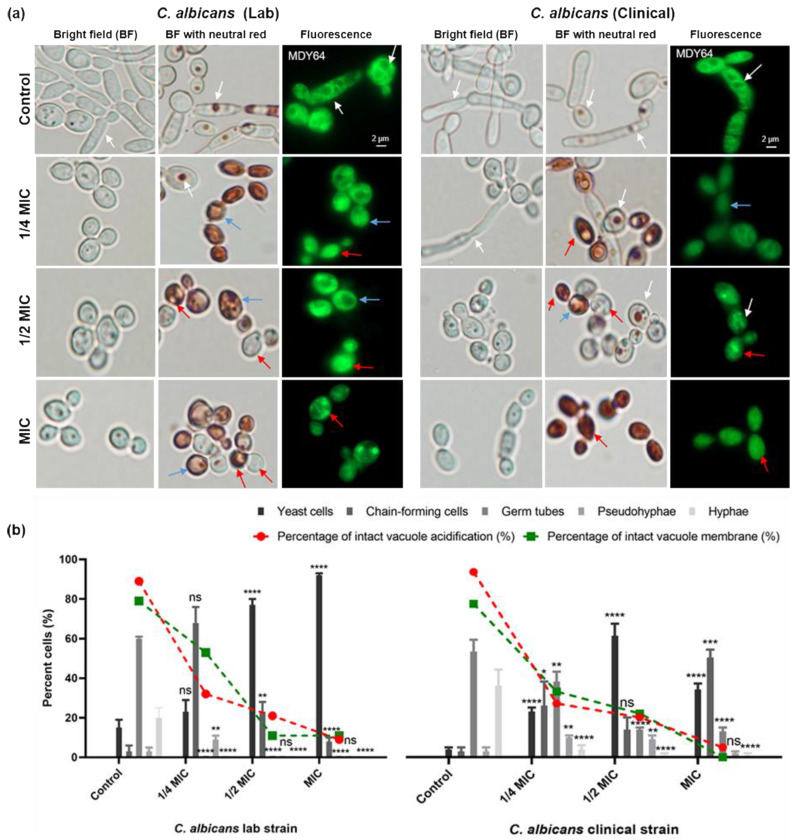
**Carvacrol inhibits germ tube formation in *C. albicans* lab and clinical strains.** (**a**) Representative bright-field and fluorescence (MDY-64: λ_ex_ = 451 nm; λ_em_ = 497 nm) microscopy images show C. albicans treated with 10% FBS in YPD medium containing carvacrol at 1/4 MIC, 1/2 MIC, and MIC (4 h exposure) form pseudohyphae and chain-forming cells with defective vacuole acidification (middle, stained with neutral red) and disrupted vacuolar membranes (MDY64-stained fluorescence images). White, blue, and red arrows indicate intact, partially perturbed, or fully perturbed vacuoles, respectively. Scale bar is 2 μm for control and applicable to all images. (**b**) Bar graphs of the percentages (%) of cells with various morphologies show a statistically significant decrease in hyphal formation for carvacrol-treated cells versus controls. Lines highlight the % cells with intact vacuoles and a statistically significant decrease in intact vacuoles in treated cells. Data are reported as means ± SEMs of 300 cells captured from ten microscopic FOVs (bright field 100×; fluorescence microscopy 63×) for three biological replicates. Statistical significance was analyzed using a one-way ANOVA, followed by Dunnett’s multiple comparison test of each condition versus control, where asterisks indicate *p*-values of <0.0001 (****), <0.001 (***), <0.01 (**), <0.05 (*), >0.05 (ns, no significance).

**Figure 5 microorganisms-11-02915-f005:**
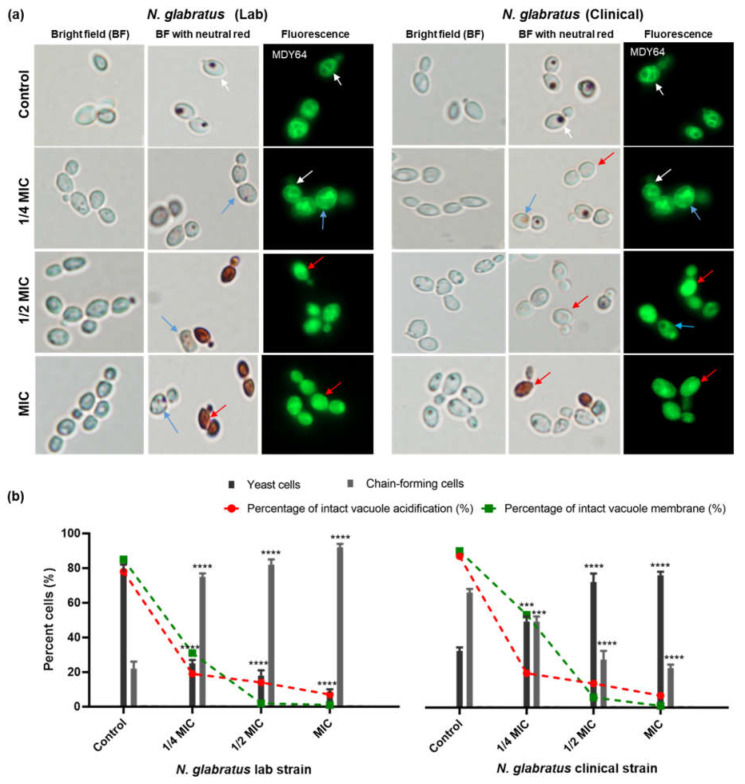
Impact of carvacrol on *N. glabratus* lab and clinical strain morphology (**a**) Representative merged bright-field and fluorescence (MDY-64: λ_ex_ = 451 nm; λ_em_ = 497 nm) microscopy images of *N. glabratus* treated with 10% FBS in YPD medium with carvacrol at 1/4, 1/2 MIC, and MIC (4 h exposure). Treated cells aggregate with acidified vacuoles (middle, stained with neutral red) and disrupted vacuolar membranes (MDY64-stained fluorescence images), while controls consist of budding yeast and healthy vacuoles. White, blue, and red arrows represent intact, partially perturbed, or fully perturbed vacuoles, respectively. Scale bar is 2 μm for control and applicable to all images. (**b**) Bar graphs of the percentages (%) of cells with various morphologies show a statistically significant increase in chain-forming cells for treated strains versus controls. The line graphs show the percentage (%) of cells with intact vacuoles, showing a statistically significant decrease in intact vacuoles for treated cells. Data are means ± SEMs of 300 cells captured from ten microscopic FOVs (bright field 100×; fluorescence microscopy 63×) for each of three biological replicates. The statistical significance was analyzed using a one-way ANOVA followed by Dunnett’s multiple comparison test of each condition versus the control, where asterisks indicate *p*-values of <0.0001 (****) and <0.001 (***).

## Data Availability

Data are contained within the article.

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
