# Peer review of "Carvacrol-Induced Vacuole Dysfunction and Morphological Consequences in *Nakaseomyces glabratus* and *Candida albicans"

_microorganisms, 2023, doi:10.3390/microorganisms11122915_

Round 1

Reviewer 1 Report

Comments and Suggestions for Authors

In the manuscript „Vacuole dysfunction and hyphal inhibition of Candida spp by carvacrol” Acuna and colleagues investigated the impact of natural product on some functionalities of Candida cells. This is an interesting work due to the constant search for new therapeutic strategies in the treatment of candidiasis, especially since the authors in their manuscript attempt to explain the mechanisms of action of carvacrol.

I have a few comments on the manuscript. The major one is that the authors almost did not emphasize how big the difference is in cell physiology, genetics and virulence of the two species they describe. Importantly, C. glabrata should now be referred to as Nakaseomyces glabratus under the latest nomenclature, due to many features that distinguish this species from the genus Candida. It is a haploid, unlike the diploid C. albicans, and, above all, it does not produce filamentous forms (hyphae or pseudohyphae). In numerous fragments in the manuscript, the text is unclear and misleading in this regard, and the filamentation of N. glabratus is suggested. This requires revisions throughout the manuscript. N. glabratus can form aggregates, but they are not called pseudohyphae. Also in Fig. 5 markings for hyphae and pseudohyphae should be removed.

Thus, the title of the manuscript should be changed to inform which two particular species were investigated.

Abstract, lines 9-10: it should be: by Candida albicans and non-albicans Candida species.

Lines 31-32 – there are several different circumstances prevailing to candidiasis, not only cancer, please mention about others.

Line 62 – it should be: Vacuoles, the largest organelle of Candida cells

Line 110 – the cell number measurements were performed by OD or CFU counting? Not all light-scattering cells may be viable.

The mechanism of filamentation depends on many different factors, not only on vacuoles, the authors should emphasize this in the text of the manuscript.

To additionally explain the mechanisms of action of carvacrol, authors should test the Ca2+ after incubation with both species, as calcium levels also influence morphological changes.

Minor: Throughout the manuscript, the names of species and genera should be written in italics (also in reference list). The name of the genus in the first capital letter, the full name at the first mention, then the abbreviation.

Comments on the Quality of English Language

Some minor mistakes in sentence syntax can be found in the manuscript text.

Author Response

I have a few comments on the manuscript. The major one is that the authors almost did not emphasize how big the difference is in cell physiology, genetics and virulence of the two species they describe. Importantly, C. glabrata should now be referred to as Nakaseomyces glabratus under the latest nomenclature, due to many features that distinguish this species from the genus Candida. It is a haploid, unlike the diploid C. albicans, and, above all, it does not produce filamentous forms (hyphae or pseudohyphae). In numerous fragments in the manuscript, the text is unclear and misleading in this regard, and the filamentation of N. glabratus is suggested. This requires revisions throughout the manuscript. N. glabratus can form aggregates, but they are not called pseudohyphae. Also in Fig. 5 markings for hyphae and pseudohyphae should be removed.

We thank the reviewer for catching this error. We have corrected the genus/species name through the manuscript, and made a clear distinction between the two strains in the introduction (lines 37-44), which is now consistent throughout.

How big the difference is in cell physiology, genetics and virulence of the two species they describe.

We now provide a comparison of the cell physiology and virulence in the introduction (lines 37-44).

Also in Fig. 5 markings for hyphae and pseudohyphae should be removed.

We have corrected the figure 5 legend.

Thus, the title of the manuscript should be changed to inform which two particular species were investigated.

We have changed the title of the manuscript accordingly.

Abstract, lines 9-10: it should be: by Candida albicans and non-albicans Candida species.

This has been corrected (lines 9-10).

Lines 31-32 – there are several different circumstances prevailing to candidiasis, not only cancer, please mention about others.

Thank you for pointing out this oversight. We have added these additional circumstances to the introduction (lines 31-34).

Line 62 – it should be: Vacuoles, the largest organelle of Candida cells

This has been corrected (line 68).

Line 110 – the cell number measurements were performed by OD or CFU counting? Not all light-scattering cells may be viable.

Cells were enumerated by CFU, as mentioned in the methods section (vacuole acidification (e.g. line 115).

The mechanism of filamentation depends on many different factors, not only on vacuoles, the authors should emphasize this in the text of the manuscript.

We absolutely agree, but we do not think that we have implied that vacuolar health is the only contributor to filamentation, just one factor.

To additionally explain the mechanisms of action of carvacrol, authors should test the Ca2+ after incubation with both species, as calcium levels also influence morphological changes.

We consider the proposed experiments to be beyond the scope of this manuscript, but we will pursue them in future research.

Minor: Throughout the manuscript, the names of species and genera should be written in italics (also in reference list). The name of the genus in the first capital letter, the full name at the first mention, then the abbreviation.

We thank you for pointing out this omission. We have italicized Genus species throughout.

Some minor mistakes in sentence syntax can be found in the manuscript text.

We have made a number of revisions throughout the text to improve syntax, highlighted in red.

Reviewer 2 Report

Comments and Suggestions for Authors

Eliz Acuna and colleagues present an article on the effects of the monoterpene carvacrol in the vacuolar function of Candida species, using growth tests and microscopic analyses in the presence of appropriate dyes. Specifically, authors demonstrate that this compound is influencing the vacuolar acidification and vacuolar membrane integrity, and speculate about a possible correlation of these results with the also observed disruption in hyphal formation in C. albicans and inhibition of chain formation in cells of C. glabrata.

In general, the article is comprehensibly written and contains interesting and important information for people working in this field. What sems to be missing in my opinion is a more detailed analysis of growth inhibition in the first chapter of the results, for example by adding growth tests on petri dishes along with graphs showing growth rates and doubling times.

Line 14-15: Please avoid taking about biochemical analyses and “uncovering the mechanism”, as no such information is offered in this manuscript.

Line 280, Chapter Title: I guess the authors mean that Carvacrol “induces” chain forming cells and not “inhibits”. 

The Discussion may benefit from some shortening.

Figure S1: The yellow font of the scale bar cannot easily be distinguished. Since all bars represent 50 μm, please consider adding just one bar in one of the pictures.

Author Response

In general, the article is comprehensibly written and contains interesting and important information for people working in this field. What seems to be missing in my opinion is a more detailed analysis of growth inhibition in the first chapter of the results, for example by adding growth tests on petri dishes along with graphs showing growth rates and doubling times.

We understand the importance of checking growth rates and doubling time. However, we have not manipulated our strains by deleting genes expected to affect the growth rates. We propose that figure 1 contains the requested information, showing growth reduction as a function of carvacrol concentration. The goal of this research was to determine how vacuole defects may relate to hyphal inhibition.

Line 14-15: Please avoid taking about biochemical analyses and “uncovering the mechanism”, as no such information is offered in this manuscript.

We have removed any reference to having determined mechanisms in the manuscript.

Line 280, Chapter Title: I guess the authors mean that Carvacrol “induces” chain forming cells and not “inhibits”. 

You are correct, and we have revised this (line 305).

The Discussion may benefit from some shortening.

We have endeavored to make the discussion more concise.

Figure S1: The yellow font of the scale bar cannot easily be distinguished. Since all bars represent 50 μm, please consider adding just one bar in one of the pictures.

Thank you for pointing this out. We have corrected this in figures 2 and S1.

Reviewer 3 Report

Comments and Suggestions for Authors

Dear Authors,

The manuscript with title "Vacuole dysfunction and hyphal inhibition of Candida spp by carvacrol" is very interested and deals with new, important and scientifically high valuable data on a significant reduction in survival of candidiasis after exposure to carvacrol.

The findings reveal that carvacrol demonstrates a broad spectrum of antifungal effects, encompassing the inhibition of growth, hyphal formation, and biofilm development. Through a combination of biochemical analyses and microscopic observations, the study elucidates the mechanism behind carvacrol-induced hyphal inhibition. Notably, exposure to carvacrol leads to a substantial reduction in candida survival rates, coupled with dose-dependent vacuolar acidification, compromised membrane integrity, and irregular morphology.

However, I will ask for one minor suggestion. On Figures, for example fig.4 and 5, it is difficult to understand which color corresponds to which column in the diagram. Please correct the colors.

Author Response

The findings reveal that carvacrol demonstrates a broad spectrum of antifungal effects, encompassing the inhibition of growth, hyphal formation, and biofilm development. Through a combination of biochemical analyses and microscopic observations, the study elucidates the mechanism behind carvacrol-induced hyphal inhibition. Notably, exposure to carvacrol leads to a substantial reduction in candida survival rates, coupled with dose-dependent vacuolar acidification, compromised membrane integrity, and irregular morphology.

We thank you for your kind review.

However, I will ask for one minor suggestion. On Figures, for example fig.4 and 5, it is difficult to understand which color corresponds to which column in the diagram. Please correct the colors.

We are not sure if you are referring to the legend of cell type (greyscale), points on the graph (red/green), denoted in the legend, or other. The meaning of arrow colour is clearly specified in the figure legends: “The white, blue and red arrows are intact, partially or fully perturbed vacuoles, respectively.”

Round 2

Reviewer 1 Report

Comments and Suggestions for Authors

Authors addressed almost all comments. However, I still insist that the authors emphasize in the text the fact that the mechanism of filamentation depends on many different factors, not only on vacuoles, and mention them briefly.

Author Response

Authors addressed almost all comments.

Thank you.

However, I still insist that the authors emphasize in the text the fact that the mechanism of filamentation depends on many different factors, not only on vacuoles, and mention them briefly.

We have added the following to the introduction (lines 46-49):

For example, the host tissue environment, including temperature (37 °C), serum, neutral pH and hormones, can induced C.albicans’ yeast-hyphal morphological switching [2,3], considered a virulence trait along with ...

Reviewer 2 Report

Comments and Suggestions for Authors

The revised manuscript has been improved. In terms of highlighting novelty however, I still think that this work would have benefited from some more detailed analysis of growth inhibition in the presence of carvacrol. Afterall the authors state in Lines 324-325 that their results on carvacrol are consistent with previously published data [21,42–44]. Other previous publications have also studied carvacrol toxicity in Candida species (e.g. doi: 10.1007/s00203-022-03355-1; doi: 10.1080/14786410903565184; doi: 10.1016/j.ijantimicag.2008.01.028; doi: 10.1016/j.resmic.2021.103916). Authors may want to cite and discuss some of these or other similar articles. I do understand that the authors do not wish to engage in additional experiments and I leave such a decision up to the editor. I have no further comments for the authors.

Author Response

The revised manuscript has been improved.

Thank you.

In terms of highlighting novelty however, I still think that this work would have benefited from some more detailed analysis of growth inhibition in the presence of carvacrol. Afterall the authors state in Lines 324-325 that their results on carvacrol are consistent with previously published data [21,42–44].

The manuscript includes a MIC assay for all strains, really simply to confirm the impact of carvacrol for comparison with prior literature, as a bench mark. However, given the prior studies of carvacrol MIC, we did not think that this aspect was particularly novel, which is why we focused on the vacuolar impacts.

Other previous publications have also studied carvacrol toxicity in Candida species (e.g. doi: 10.1007/s00203-022-03355-1; doi: 10.1080/14786410903565184; doi: 10.1016/j.ijantimicag.2008.01.028; doi: 10.1016/j.resmic.2021.103916). Authors may want to cite and discuss some of these or other similar articles. 

Is the reviewer's issue that we have overlooked references? It is always difficult to strike a balance between capturing everything in the literature, and not over referencing. While we considered the references suggested by the reviewer, we thought that there were other references that better captured the essence of the study:

  1. "In vivo" and "in vitro" antimicrobial activity of Origanum vulgare essential oil and its two phenolic compounds on clinical isolates of Candida spp (2022) - while this is an excellent paper, and my lab has used both confocal and SEM to study the impact of essential oil components on Candida, that was not the focus of this study. For this reason, we did not cite this article that examines the impact of carvacrol on the cell wall.
  2. Evaluation of in vitro activity of carvacrol against Candida albicans strains  (2010) - our major focus was not MIC, so we did not include this reference, rather more recent ones that also include MIC values

  3. In vitro activity of terpenes against Candida biofilms (2008) - instead of this article that looks at the impact of carvacrol on C. albicans biofilms, we included a more recent publication with a similar focus: Miranda-Cadena, K.; Marcos-Arias, C.; Mateo.; Aguirre-Urizar, J.M.; Quindós G, Eraso E. In vitro activities of carvacrol, cinnamaldehyde and thymol against Candida biofilms. Biomed. Pharmacother. 2021,143; https://doi.org/10.1016/j.biopha.2021.112218. 

  4. Carvacrol modulates the expression and activity of antioxidant enzymes in Candida auris (2022) - it is generally well accepted that most essential oil components induce oxidative stress. That said, in this study we did not examine the oxidative impact of carvacrol.